# MOGRIFIER LSTM

**Gábor Melis[†], Tomáš Kočiský[†], Phil Blunsom[†‡]**
{melisgl,tkocisky,pblunsom}@google.com
[†]DeepMind, London, UK
[‡]University of Oxford

## ABSTRACT

Many advances in Natural Language Processing have been based upon more expressive models for how inputs interact with the context in which they occur. Recurrent networks, which have enjoyed a modicum of success, still lack the generalization and systematicity ultimately required for modelling language. In this work, we propose an extension to the venerable Long Short-Term Memory in the form of mutual gating of the current input and the previous output. This mechanism affords the modelling of a richer space of interactions between inputs and their context. Equivalently, our model can be viewed as making the transition function given by the LSTM context-dependent. Experiments demonstrate markedly improved generalization on language modelling in the range of 3–4 perplexity points on Penn Treebank and Wikitext-2, and 0.01–0.05 bpc on four character-based datasets. We establish a new state of the art on all datasets with the exception of Enwik8, where we close a large gap between the LSTM and Transformer models.

## 1 INTRODUCTION

The domination of Natural Language Processing by neural models is hampered only by their limited ability to generalize and questionable sample complexity (Belinkov and Bisk 2017; Jia and Liang 2017; Iyyer et al. 2018; Moosavi and Strube 2017; Agrawal et al. 2016), their poor grasp of grammar (Linzen et al. 2016; Kuncoro et al. 2018), and their inability to chunk input sequences into meaningful units (Wang et al. 2017). While direct attacks on the latter are possible, in this paper, we take a language-agnostic approach to improving Recurrent Neural Networks (RNN, Rumelhart et al. (1988)), which brought about many advances in tasks such as language modelling, semantic parsing, machine translation, with no shortage of non-NLP applications either (Bakker 2002; Mayer et al. 2008). Many neural models are built from RNNs including the sequence-to-sequence family (Sutskever et al. 2014) and its attention-based branch (Bahdanau et al. 2014). Thus, innovations in RNN architecture tend to have a trickle-down effect from language modelling, where evaluation is often the easiest and data the most readily available, to many other tasks, a trend greatly strengthened by ULMFiT (Howard and Ruder 2018), ELMo (Peters et al. 2018) and BERT (Devlin et al. 2018), which promote language models from architectural blueprints to pretrained building blocks.

To improve the generalization ability of language models, we propose an extension to the LSTM (Hochreiter and Schmidhuber 1997), where the LSTM's input $x$ is gated conditioned on the output of the previous step $h_{prev}$. Next, the gated input is used in a similar manner to gate the output of the previous time step. After a couple of rounds of this mutual gating, the last updated $x$ and $h_{prev}$ are fed to an LSTM. By introducing these additional of gating operations, in one sense, our model joins the long list of recurrent architectures with gating structures of varying complexity which followed the invention of Elman Networks (Elman 1990). Examples include the LSTM, the GRU (Chung et al. 2015), and even designs by Neural Architecture Search (Zoph and Le 2016).

Intuitively, in the lowermost layer, the first gating step scales the input embedding (itself a representation of the *average* context in which the token occurs) depending on the *actual* context, resulting in a contextualized representation of the input. While intuitive, as Section 4 shows, this interpretation cannot account for all the observed phenomena.

In a more encompassing view, our model can be seen as enriching the mostly additive dynamics of recurrent transitions placing it in the company of the Input Switched Affine Network (Foerster et al.

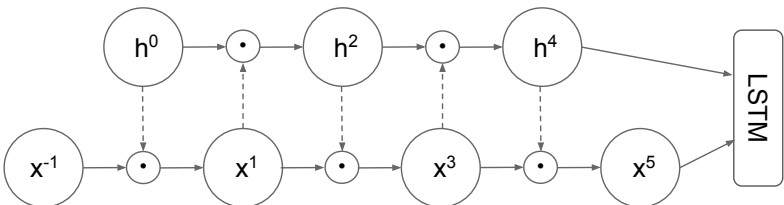

Figure 1: Mogrifier with 5 rounds of updates. The previous state $\boldsymbol{h}^0 = \boldsymbol{h}_{prev}$ is transformed linearly (dashed arrows), fed through a sigmoid and gates $\boldsymbol{x}^{-1} = \boldsymbol{x}$ in an elementwise manner producing $\boldsymbol{x}^1$. Conversely, the linearly transformed $\boldsymbol{x}^1$ gates $\boldsymbol{h}^0$ and produces $\boldsymbol{h}^2$. After a number of repetitions of this mutual gating cycle, the last values of $\boldsymbol{h}^*$ and $\boldsymbol{x}^*$ sequences are fed to an LSTM cell. The *prev* subscript of $\boldsymbol{h}$ is omitted to reduce clutter.

2017) with a separate transition matrix for each possible input, and the Multiplicative RNN (Sutskever et al. 2011), which factorizes the three-way tensor of stacked transition matrices. Also following this line of research are the Multiplicative Integration LSTM (Wu et al. 2016) and – closest to our model in the literature – the Multiplicative LSTM (Krause et al. 2016). The results in Section 3.4 demonstrate the utility of our approach, which consistently improves on the LSTM and establishes a new state of the art on all but the largest dataset, Enwik8, where we match similarly sized transformer models.

## 2 MODEL

To allow for ease of subsequent extension, we present the standard LSTM update (Sak et al. 2014) with input and state of size $m$ and $n$ respectively as the following function:

$$\mathrm{LSTM}\colon \mathbb{R}^m \times \mathbb{R}^n \times \mathbb{R}^n \to \mathbb{R}^n \times \mathbb{R}^n$$
$$\mathrm{LSTM}(\boldsymbol{x}, \boldsymbol{c}_{prev}, \boldsymbol{h}_{prev}) = (\boldsymbol{c}, \boldsymbol{h}).$$

The updated state $\boldsymbol{c}$ and the output $\boldsymbol{h}$ are computed as follows:

$$\boldsymbol{f} = \sigma(\mathbf{W}^{fx}\boldsymbol{x} + \mathbf{W}^{fh}\boldsymbol{h}_{prev} + \boldsymbol{b}^f)$$
$$\boldsymbol{i} = \sigma(\mathbf{W}^{ix}\boldsymbol{x} + \mathbf{W}^{ih}\boldsymbol{h}_{prev} + \boldsymbol{b}^i)$$
$$\boldsymbol{j} = \tanh(\mathbf{W}^{jx}\boldsymbol{x} + \mathbf{W}^{jh}\boldsymbol{h}_{prev} + \boldsymbol{b}^j)$$
$$\boldsymbol{o} = \sigma(\mathbf{W}^{ox}\boldsymbol{x} + \mathbf{W}^{oh}\boldsymbol{h}_{prev} + \boldsymbol{b}^o)$$
$$\boldsymbol{c} = \boldsymbol{f} \odot \boldsymbol{c}_{prev} + \boldsymbol{i} \odot \boldsymbol{j}$$
$$\boldsymbol{h} = \boldsymbol{o} \odot \tanh(\boldsymbol{c}),$$

where $\sigma$ is the logistic sigmoid function, $\odot$ is the elementwise product, $\mathbf{W}^{**}$ and $b^*$ are weight matrices and biases.

While the LSTM is typically presented as a solution to the vanishing gradients problem, its gate $i$ can also be interpreted as scaling the rows of weight matrices $\mathbf{W}^{j*}$ (ignoring the non-linearity in $j$). In this sense, the LSTM nudges Elman Networks towards context-dependent transitions and the extreme case of Input Switched Affine Networks. If we took another, larger step towards that extreme, we could end up with Hypernetworks (Ha et al. 2016). Here, instead, we take a more cautious step, and equip the LSTM with gates that scale the *columns* of all its weight matrices $\mathbf{W}^{**}$ in a context-dependent manner. The scaling of the matrices $\mathbf{W}^{*x}$ (those that transform the cell input) makes the input embeddings dependent on the cell state, while the scaling of $\mathbf{W}^{*h}$ does the reverse.

The Mogrifier[1] LSTM is an LSTM where two inputs $\boldsymbol{x}$ and $\boldsymbol{h}_{prev}$ modulate one another in an alternating fashion before the usual LSTM computation takes place (see Fig. 1). That is, $\mathrm{Mogrify}(\boldsymbol{x}, \boldsymbol{c}_{prev}, \boldsymbol{h}_{prev}) = \mathrm{LSTM}(\boldsymbol{x}^{\uparrow}, \boldsymbol{c}_{prev}, \boldsymbol{h}^{\uparrow}_{prev})$ where the modulated inputs $\boldsymbol{x}^{\uparrow}$ and $\boldsymbol{h}^{\uparrow}_{prev}$ are defined as the highest indexed $\boldsymbol{x}^i$ and $\boldsymbol{h}^i_{prev}$, respectively, from the interleaved sequences

$$\boldsymbol{x}^i = 2\sigma(\mathbf{Q}^i\boldsymbol{h}^{i-1}_{prev}) \odot \boldsymbol{x}^{i-2}, \qquad\qquad \text{for odd } i \in [1 \ldots r] \qquad\qquad (1)$$

---

[1]It's like a transmogrifier[2] without the magic: it can only shrink or expand objects.

[2]Transmogrify (verb, 1650s): to completely alter the form of something in a surprising or magical manner.

$$\boldsymbol{h}_{prev}^{i} = 2\sigma(\mathbf{R}^{i}\boldsymbol{x}^{i-1}) \odot \boldsymbol{h}_{prev}^{i-2}, \qquad\qquad \text{for even i} \in [1\dots r] \qquad (2)$$

with $\boldsymbol{x}^{-1} = \boldsymbol{x}$ and $\boldsymbol{h}_{prev}^{0} = \boldsymbol{h}_{prev}$. The number of "rounds", $r \in \mathbb{N}$, is a hyperparameter; $r = 0$ recovers the LSTM. Multiplication with the constant 2 ensures that randomly initialized $\mathbf{Q}^{i}, \mathbf{R}^{i}$ matrices result in transformations close to identity. To reduce the number of additional model parameters, we typically factorize the $\mathbf{Q}^{i}, \mathbf{R}^{i}$ matrices as products of low-rank matrices: $\mathbf{Q}^{i} = \mathbf{Q}_{\text{left}}^{i}\mathbf{Q}_{\text{right}}^{i}$ with $\mathbf{Q}^{i} \in \mathbb{R}^{m \times n}, \mathbf{Q}_{\text{left}}^{i} \in \mathbb{R}^{m \times k}, \mathbf{Q}_{\text{right}}^{i} \in \mathbb{R}^{k \times n}$, where $k < min(m, n)$ is the rank.

## 3 EXPERIMENTS

### 3.1 THE CASE FOR SMALL-SCALE

Before describing the details of the data, the experimental setup and the results, we take a short detour to motivate work on smaller-scale datasets. A recurring theme in the history of sequence models is that the problem of model design is intermingled with optimizability and scalability. Elman Networks are notoriously difficult to optimize, a property that ultimately gave birth to the idea of the LSTM, but also to more recent models such as the Unitary Evolution RNN (Arjovsky et al. 2016) and fixes like gradient clipping (Pascanu et al. 2013). Still, it is far from clear – if we could optimize these models well – how different their biases would turn out to be. The non-separability of model and optimization is fairly evident in these cases.

Scalability, on the other hand, is often optimized for indirectly. Given the limited ability of current models to generalize, we often compensate by throwing more data at the problem. To fit a larger dataset, model size must be increased. Thus the best performing models are evaluated based on their scalability[3]. Today, scaling up still yields tangible gains on down-stream tasks, and language modelling data is abundant. However, we believe that simply scaling up will not solve the generalization problem and better models will be needed. Our hope is that by choosing small enough datasets, so that model size is no longer the limiting factor, we get a number of practical advantages:

* Generalization ability will be more clearly reflected in evaluations even without domain adaptation.
* Turnaround time in experiments will be reduced, and the freed up computational budget can be put to good use by controlling for nuisance factors.
* The transient effects of changing hardware performance characteristics are somewhat lessened.

Thus, we develop, analyse and evaluate models primarily on small datasets. Evaluation on larger datasets is included to learn more about the models' scaling behaviour and because of its relevance for applications, but it is to be understood that these evaluations come with much larger error bars and provide more limited guidance for further research on better models.

### 3.2 DATASETS

We compare models on both word and character-level language modelling datasets. The two word-level datasets we picked are the Penn Treebank (PTB) corpus by Marcus et al. (1993) with preprocessing from Mikolov et al. (2010) and Wikitext-2 by Merity et al. (2016), which is about twice the size of PTB with a larger vocabulary and lighter preprocessing. These datasets are definitely on the small side, but – and *because* of this – they are suitable for exploring different model biases. Their main shortcoming is the small vocabulary size, only in the tens of thousands, which makes them inappropriate for exploring the behaviour of the long tail. For that, open vocabulary language modelling and byte pair encoding (Sennrich et al. 2015) would be an obvious choice. Still, our primary goal here is the comparison of the LSTM and Mogrifier architectures, thus we instead opt for character-based language modelling tasks, where vocabulary size is not an issue, the long tail is not truncated, and there are no additional hyperparameters as in byte pair encoding that make fair comparison harder. The first character-based corpus is Enwik8 from the Hutter Prize dataset (Hutter 2012). Following common practice, we use the first 90 million characters for training and the remaining 10 million evenly split between validation and test. The character-level task on the

---

[3]Note that the focus on scalability is *not* a problem per se. Indeed the unsupervised pretraining methods (Peters et al. 2018; Devlin et al. 2018) take great advantage of this approach.

Table 1: Word-level perplexities of near state-of-the-art models, our **LSTM** baseline and the **Mogrifier** on PTB and Wikitext-2. Models with Mixture of Softmaxes (Yang et al. 2017) are denoted with *MoS*, depth N with *dN*. *MC* stands for Monte-Carlo dropout evaluation. Previous state-of-the-art results in italics. Note the comfortable margin of 2.8–4.3 perplexity points the Mogrifier enjoys over the LSTM.

| | | | No Dyneval | | Dyneval | |
| | | | Val. | Test | Val. | Test |
|---|---|---|---|---|---|---|
| PTB EN | FRAGE (d3, MoS15) (Gong et al. 2018) | 22M | *54.1* | *52.4* | *47.4* | *46.5* |
| | AWD-LSTM (d3, MoS15) (Yang et al. 2017) | 22M | 56.5 | 54.4 | 48.3 | 47.7 |
| | Transformer-XL (Dai et al. 2019) | 24M | 56.7 | 54.5 | | |
| | **LSTM** (d2) | 24M | 55.8 | 54.6 | 48.9 | 48.4 |
| | **Mogrifier** (d2) | 24M | 52.1 | 51.0 | 45.1 | 45.0 |
| | **LSTM** (d2, MC) | 24M | 55.5 | 54.1 | 48.6 | 48.4 |
| | **Mogrifier** (d2, MC) | 24M | **51.4** | **50.1** | **44.9** | **44.8** |
| WT2 EN | FRAGE (d3, MoS15) (Gong et al. 2018) | 35M | *60.3* | *58.0* | *40.8* | *39.1* |
| | AWD-LSTM (d3, MoS15) (Yang et al. 2017) | 35M | 63.9 | 61.2 | 42.4 | 40.7 |
| | **LSTM** (d2, MoS2) | 35M | 62.6 | 60.1 | 43.2 | 41.5 |
| | **Mogrifier** (d2, MoS2) | 35M | 58.7 | 56.6 | 40.6 | 39.0 |
| | **LSTM** (d2, MoS2, MC) | 35M | 61.9 | 59.4 | 43.2 | 41.4 |
| | **Mogrifier** (d2, MoS2, MC) | 35M | **57.3** | **55.1** | **40.2** | **38.6** |

Mikolov preprocessed PTB corpus (Merity et al. 2018) is unique in that it has the disadvantages of closed vocabulary without the advantages of word-level modelling, but we include it for comparison to previous work. The final character-level dataset is the Multilingual Wikipedia Corpus (MWC, Kawakami et al. (2017)), from which we focus on the English and Finnish language subdatasets in the single text, large setting.

## 3.3 SETUP

We tune hyperparameters following the experimental setup of Melis et al. (2018) using a black-box hyperparameter tuner based on batched Gaussian Process Bandits (Golovin et al. 2017). For the LSTM, the tuned hyperparameters are the same: *input_embedding_ratio*, *learning_rate*, *l2_penalty*, *input_dropout*, *inter_layer_dropout*, *state_dropout*, *output_dropout*. For the Mogrifier, the number of rounds $r$ and the rank $k$ of the low-rank approximation is also tuned (allowing for full rank, too). For word-level tasks, BPTT (Werbos et al. 1990) window size is set to 70 and batch size to 64. For character-level tasks, BPTT window size is set to 150 and batch size to 128 except for Enwik8 where the window size is 500. Input and output embeddings are tied for word-level tasks following Inan et al. (2016) and Press and Wolf (2016). Optimization is performed with Adam (Kingma and Ba 2014) with $\beta_1 = 0$, a setting that resembles RMSProp without momentum. Gradients are clipped (Pascanu et al. 2013) to norm 10. We switch to averaging weights similarly to Merity et al. (2017) after a certain number of checkpoints with no improvement in validation cross-entropy or at 80% of the training time at the latest. We found no benefit to using two-step finetuning.

Model evaluation is performed with the standard, deterministic dropout approximation or Monte-Carlo averaging (Gal and Ghahramani 2016) where explicitly noted (MC). In standard dropout evaluation, dropout is turned off while in MC dropout predictions are averaged over randomly sampled dropout masks (200 in our experiments). Optimal softmax temperature is determined on the validation set, and in the MC case dropout rates are scaled (Melis et al. 2018). Finally, we report results with and without dynamic evaluation (Krause et al. 2017). Hyperparameters for dynamic evaluation are tuned using the same method (see Appendix A for details).

We make the code and the tuner output available at https://github.com/deepmind/lamb.

## 3.4 RESULTS

Table 1 lists our results on word-level datasets. On the PTB and Wikitext-2 datasets, the Mogrifier has lower perplexity than the LSTM by 3–4 perplexity points regardless of whether or not dynamic evaluation (Krause et al. 2017) and Monte-Carlo averaging are used. On both datasets, the state of the art is held by the AWD LSTM (Merity et al. 2017) extended with Mixture of Softmaxes (Yang

Table 2: Bits per character on character-based datasets of near state-of-the-art models, our **LSTM** baseline and the **Mogrifier**. Previous state-of-the-art results in italics. Depth N is denoted with *dN*. MC stands for Monte-Carlo dropout evaluation. Once again the Mogrifier strictly dominates the LSTM and sets a new state of the art on all but the Enwik8 dataset where with dynamic evaluation it closes the gap to the Transformer-XL of similar size († Krause et al. (2019), ‡ Ben Krause, personal communications, May 17, 2019). On most datasets, model size was set large enough for underfitting not to be an issue. This was very much not the case with Enwik8, so we grouped models of similar sizes together for ease of comparison. Unfortunately, a couple of dynamic evaluation test runs diverged (NaN) on the test set and some were just too expensive to run (Enwik8, MC).

| | | | No Dyneval | | Dyneval | |
| | | | Val. | Test | Val. | Test |
|---|---|---|---|---|---|---|
| PTB EN | Trellis Networks (Bai et al. 2018) | 13.4M | | *1.159* | | |
| | AWD-LSTM (d3) (Merity et al. 2017) | 13.8M | | 1.175 | | |
| | **LSTM** (d2) | 24M | 1.163 | 1.143 | 1.116 | 1.103 |
| | **Mogrifier** (d2) | 24M | 1.149 | 1.131 | 1.098 | 1.088 |
| | **LSTM** (d2, MC) | 24M | 1.159 | 1.139 | 1.115 | 1.101 |
| | **Mogrifier** (d2, MC) | 24M | **1.137** | **1.120** | **1.094** | **1.083** |
| MWC EN | HCLM with Cache (Kawakami et al. 2017) | 8M | *1.591* | *1.538* | | |
| | LSTM (d1) (Kawakami et al. 2017) | 8M | 1.793 | 1.736 | | |
| | **LSTM** (d2) | 24M | 1.353 | 1.338 | 1.239 | 1.225 |
| | **Mogrifier** (d2) | 24M | 1.319 | 1.305 | 1.202 | 1.188 |
| | **LSTM** (d2, MC) | 24M | 1.346 | 1.332 | 1.238 | NaN |
| | **Mogrifier** (d2, MC) | 24M | **1.312** | **1.298** | **1.200** | **1.187** |
| MWC FI | HCLM with Cache (Kawakami et al. 2017) | 8M | *1.754* | *1.711* | | |
| | LSTM (d1) (Kawakami et al. 2017) | 8M | 1.943 | 1.913 | | |
| | **LSTM** (d2) | 24M | 1.382 | 1.367 | 1.249 | 1.237 |
| | **Mogrifier** (d2) | 24M | 1.338 | 1.326 | 1.202 | **1.191** |
| | **LSTM** (d2, MC) | 24M | 1.377 | 1.361 | 1.247 | 1.234 |
| | **Mogrifier** (d2, MC) | 24M | **1.327** | **1.313** | **1.198** | NaN |
| Enwik8 EN | Transformer-XL (d24) (Dai et al. 2019) | 277M | | **0.993** | | **0.940**† |
| | Transformer-XL (d18) (Dai et al. 2019) | 88M | | 1.03 | | |
| | **LSTM** (d4) | 96M | 1.145 | 1.155 | 1.041 | 1.020 |
| | **Mogrifier** (d4) | 96M | 1.110 | 1.122 | 1.009 | 0.988 |
| | **LSTM** (d4, MC) | 96M | 1.139 | 1.147 | | |
| | **Mogrifier** (d4, MC) | 96M | 1.104 | 1.116 | | |
| | Transformer-XL (d12) (Dai et al. 2019) | 41M | | 1.06 | | 1.01‡ |
| | AWD-LSTM (d3) (Merity et al. 2017) | 47M | | 1.232 | | |
| | mLSTM (d1) (Krause et al. 2016) | 46M | | 1.24 | | 1.08 |
| | **LSTM** (d4) | 48M | 1.182 | 1.195 | 1.073 | 1.051 |
| | **Mogrifier** (d4) | 48M | 1.135 | 1.146 | 1.035 | 1.012 |
| | **LSTM** (d4, MC) | 48M | 1.176 | 1.188 | | |
| | **Mogrifier** (d4, MC) | 48M | 1.130 | 1.140 | | |

et al. 2017) and FRAGE (Gong et al. 2018). The Mogrifier improves the state of the art without either of these methods on PTB, and without FRAGE on Wikitext-2.

Table 2 lists the character-level modelling results. On all datasets, our baseline LSTM results are much better than those previously reported for LSTMs, highlighting the issue of scalability and experimental controls. In some cases, these unexpectedly large gaps may be down to lack of hyperparameter tuning as in the case of Merity et al. (2017), or in others, to using a BPTT window size (50) that is too small for character-level modelling (Melis et al. 2017) in order to fit the model into memory. The Mogrifier further improves on these baselines by a considerable margin. Even the smallest improvement of 0.012 bpc on the highly idiosyncratic, character-based, Mikolov preprocessed PTB task is equivalent to gaining about 3 perplexity points on word-level PTB. MWC, which was built for open-vocabulary language modelling, is a much better smaller-scale character-level dataset. On the English and the Finnish corpora in MWC, the Mogrifier enjoys a gap of 0.033-0.046 bpc. Finally, on the Enwik8 dataset, the gap is 0.029-0.039 bpc in favour of the Mogrifier.

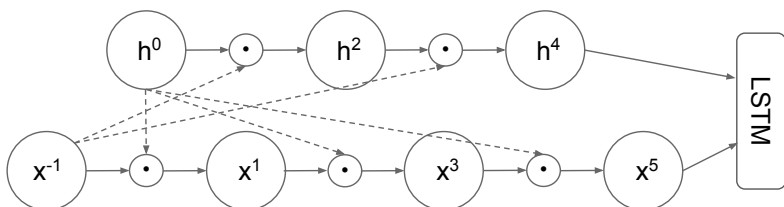

Figure 2: "No-zigzag" Mogrifier for the ablation study. Gating is always based on the original inputs.

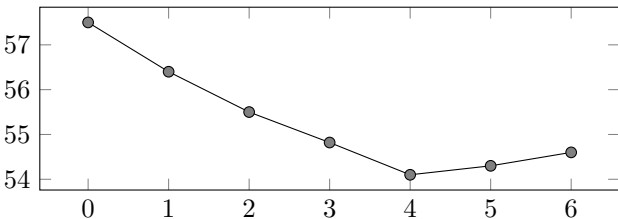

Figure 3: Perplexity vs the rounds $r$ in the PTB ablation study.

Table 3: PTB ablation study validation perplexities with 24M parameters.

| | |
|---|---|
| Mogrifier | 54.1 |
| Full rank $Q^i, P^i$ | 54.6 |
| No zigzag | 55.0 |
| LSTM | 57.5 |
| mLSTM | 57.8 |

Of particular note is the comparison to Transformer-XL (Dai et al. 2019), a state-of-the-art model on larger datasets such as Wikitext-103 and Enwik8. On PTB, without dynamic evaluation, the Transformer-XL is on par with our LSTM baseline which puts it about 3.5 perplexity points behind the Mogrifier. On Enwik8, also without dynamic evaluation, the Transformer-XL has a large, 0.09 bpc advantage at similar parameter budgets, but with dynamic evaluation this gap disappears. However, we did not test the Transformer-XL ourselves, so fair comparison is not possible due to differing experimental setups and the rather sparse result matrix for the Transformer-XL.

## 4 ANALYSIS

### 4.1 ABLATION STUDY

The Mogrifier consistently outperformed the LSTM in our experiments. The optimal settings were similar across all datasets, with $r \in \{5, 6\}$ and $k \in [40 \ldots 90]$ (see Appendix B for a discussion of hyperparameter sensitivity). In this section, we explore the effect of these hyperparameters and show that the proposed model is not unnecessarily complicated. To save computation, we tune all models using a shortened schedule with only 145 epochs instead of 964 and a truncated BPTT window size of 35 on the word-level PTB dataset, and evaluate using the standard, deterministic dropout approximation with a tuned softmax temperature.

Fig. 3 shows that the number of rounds $r$ greatly influences the results. Second, we found the low-rank factorization of $\mathbf{Q}^i$ and $\mathbf{R}^i$ to help a bit, but the full-rank variant is close behind which is what we observed on other datasets, as well. Finally, to verify that the alternating gating scheme is not overly complicated, we condition *all* newly introduced gates on the original inputs $\boldsymbol{x}$ and $\boldsymbol{h}_{prev}$ (see Fig. 2). That is, instead of Eq. 1 and Eq. 2 the no-zigzag updates are

$$\boldsymbol{x}^i = 2\sigma(\mathbf{Q}^i \boldsymbol{h}_{prev}) \odot \boldsymbol{x}^{i-2} \qquad \text{for odd i} \in [1 \ldots r],$$
$$\boldsymbol{h}_{prev}^i = 2\sigma(\mathbf{R}^i \boldsymbol{x}) \odot \boldsymbol{h}_{prev}^{i-2} \qquad \text{for even i} \in [1 \ldots r].$$

In our experiments, the no-zigzag variant underperformed the baseline Mogrifier by a small but significant margin, and was on par with the $r = 2$ model in Fig. 3 suggesting that the Mogrifier's iterative refinement scheme does more than simply widen the range of possible gating values of $\boldsymbol{x}$ and $\boldsymbol{h}_{prev}$ to $(0, 2^{\lceil r/2 \rceil})$ and $(0, 2^{\lfloor r/2 \rfloor})$, respectively.

### 4.2 COMPARISON TO THE MLSTM

The Multiplicative LSTM (Krause et al. 2016), or mLSTM for short, is closest to our model in the literature. It is defined as $\text{mLSTM}(\boldsymbol{x}, \boldsymbol{c}_{prev}, \boldsymbol{h}_{prev}) = \text{LSTM}(\boldsymbol{x}, \boldsymbol{c}_{prev}, \boldsymbol{h}_{prev}^m)$, where $\boldsymbol{h}_{prev}^m =$

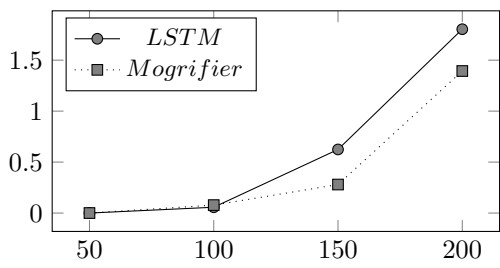 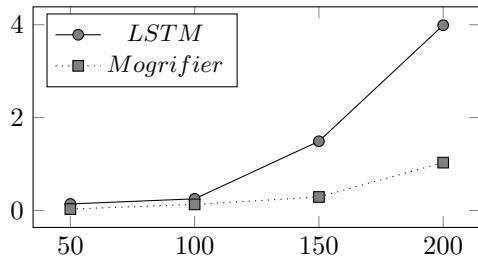

(a) 10M model parameters with vocabulary size 1k.   (b) 24M model parameters with vocabulary size 10k.

Figure 4: Cross-entropy vs sequence length in the reverse copy task with i.i.d. tokens. Lower is better. The Mogrifier is better than the LSTM even in this synthetic task with no resemblance to natural language.

$(\mathbf{W}^{mx}\boldsymbol{x}) \odot (\mathbf{W}^{mh}\boldsymbol{h}_{prev})$. In this formulation, the differences are readily apparent. First, the mLSTM allows for multiplicative interaction between $\boldsymbol{x}$ and $\boldsymbol{h}_{prev}$, but it only overrides $\boldsymbol{h}_{prev}$, while in the Mogrifier the interaction is two-way, which – as the ablation study showed – is important. Second, the mLSTM can change not only the magnitude but also the sign of values in $\boldsymbol{h}_{prev}$, something with which we experimented in the Mogrifier, but could not get to work. Furthermore, in the definition of $\boldsymbol{h}_{prev}^{m}$, the unsquashed linearities and their elementwise product make the mLSTM more sensitive to initialization and unstable during optimization.

On the Enwik8 dataset, we greatly improved on the published results of the mLSTM (Krause et al. 2016). In fact, even our LSTM baseline outperformed the mLSTM by 0.03 bpc. We also conducted experiments on PTB based on our reimplementation of the mLSTM following the same methodology as the ablation study and found that the mLSTM did not improve on the LSTM (see Table 3).

Krause et al. (2016) posit and verify the recovery hypothesis which says that having just suffered a large loss, the loss on the next time step will be smaller on average for the mLSTM than for the LSTM. This was found not to be the case for the Mogrifier. Neither did we observe a significant change in the gap between the LSTM and the Mogrifier in the tied and untied embeddings settings, which would be expected if recovery was affected by $\boldsymbol{x}$ and $\boldsymbol{h}_{prev}$ being in different domains.

### 4.3   THE REVERSE COPY TASK

Our original motivation for the Mogrifier was to allow the context to amplify salient and attenuate nuisance features in the input embeddings. We conduct a simple experiment to support this point of view. Consider the reverse copy task where the network reads an input sequence of tokens and a marker token after which it has to repeat the input in reverse order. In this simple sequence-to-sequence learning (Sutskever et al. 2014) setup, the reversal is intended to avoid the minimal time lag problem (Hochreiter and Schmidhuber 1997), which is not our focus here.

The experimental setup is as follows. For the training set, we generate 500 000 examples by uniformly sampling a given number of tokens from a vocabulary of size 1000. The validation and test sets are constructed similarly, and contain 10 000 examples. The model consists of an independent, unidirectional encoder and a decoder, whose total number of parameters is 10 million. The decoder is initialized from the last state of the encoder. Since overfitting is not an issue here, no dropout is necessary, and we only tune the learning rate, the l2 penalty, and the embedding size for the LSTM. For the Mogrifier, the number of rounds $r$ and the rank $k$ of the low-rank approximation are also tuned.

We compare the case where both the encoder and decoder are LSTMs to where both are Mogrifiers. Fig. 4a shows that, for sequences of length 50 and 100, both models can solve the task perfectly. At higher lengths though, the Mogrifier has a considerable advantage. Examining the best hyperparameter settings found, the embedding/hidden sizes for the LSTM and Mogrifier are 498/787 vs 41/1054 at 150 steps, and 493/790 vs 181/961 at 200 steps. Clearly, the Mogrifier was able to work with a much smaller embedding size than the LSTM, which is in line with our expectations for a model with a more flexible interaction between the input and recurrent state. We also conducted experiments with a larger model and vocabulary size, and found the effect even more pronounced (see Fig. 4b).

### 4.4 WHAT THE MOGRIFIER IS NOT

The results on the reverse copy task support our hypothesis that input embeddings are enriched by the Mogrifier architecture, but that cannot be the full explanation as the results of the ablation study indicate. In the following, we consider a number of hypotheses about where the advantage of the Mogrifier lies and the experiments that provide evidence *against* them.

- ♯ *Hypothesis: the benefit is in scaling $x$ and $h_{prev}$.* We verified that data dependency is a crucial feature by adding a learnable scaling factor to the LSTM inputs. We observed no improvement. Also, at extremely low-rank (less than 5) settings where the amount of information in its gating is small, the Mogrifier loses its advantage.

- ♯ *Hypothesis: the benefit is in making optimization easier.* We performed experiments with different optimizers (SGD, RMSProp), with intra-layer batch normalization and layer normalization on the LSTM gates. While we cannot rule out an effect on optimization difficulty, in all of these experiments the gap between the LSTM and the Mogrifier was the same.

- ♯ *Hypothesis: exact tying of embeddings is too constraining, the benefit is in making this relationship less strict.* Experiments conducted with untied embeddings and character-based models demonstrate improvements of similar magnitude.

- ♯ *Hypothesis: the benefit is in the low-rank factorization of $\mathbf{Q}^i, \mathbf{R}^i$ implicitly imposing structure on the LSTM weight matrices.* We observed that the full-rank Mogrifier also performed better than the plain LSTM. We conducted additional experiments where the LSTM's gate matrices were factorized and observed no improvement.

- ♯ *Hypothesis: the benefit comes from better performance on rare words.* The observed advantage on character-based modelling is harder to explain based on frequency. Also, in the reverse copy experiments, a large number of tokens were sampled uniformly, so there were no rare words at all.

- ♯ *Hypothesis: the benefit is specific to the English language.* This is directly contradicted by the Finnish MWC and the reverse copy experiments.

- ♯ *Hypothesis: the benefit is in handling long-range dependencies better.* Experiments in the episodic setting (i.e. sentence-level language modelling) exhibited the same gap as the non-episodic ones.

- ♯ *Hypothesis: the scaling up of inputs saturates the downstream LSTM gates.* The idea here is that saturated gates may make states more stable over time. We observed the opposite: the means of the standard LSTM gates in the Mogrifier were very close between the two models, but their variance was smaller in the Mogrifier.

## 5 CONCLUSIONS AND FUTURE WORK

We presented the Mogrifier LSTM, an extension to the LSTM, with state-of-the-art results on several language modelling tasks. Our original motivation for this work was that the context-free representation of input tokens may be a bottleneck in language models and by conditioning the input embedding on the recurrent state some benefit was indeed derived. While it may be part of the explanation, this interpretation clearly does not account for the improvements brought by conditioning the recurrent state on the input and especially the applicability to character-level datasets. Positioning our work on the Multiplicative RNN line of research offers a more compelling perspective.

To give more credence to this interpretation, in the analysis we highlighted a number of possible alternative explanations, and ruled them all out to varying degrees. In particular, the connection to the mLSTM is weaker than expected as the Mogrifier does not exhibit improved recovery (see Section 4.2), and on PTB the mLSTM works only as well as the LSTM. At the same time, the evidence against easier optimization is weak, and the Mogrifier establishing some kind of sharing between otherwise independent LSTM weight matrices is a distinct possibility.

Finally, note that as shown by Fig. 1 and Eq. 1-2, the Mogrifier is a series of preprocessing steps composed with the LSTM function, but other architectures, such as Mogrifier GRU or Mogrifier Elman Network are possible. We also leave investigations into other forms of parameterization of context-dependent transitions for future work.

## ACKNOWLEDGMENTS

We would like to thank Ben Krause for the Transformer-XL dynamic evaluation results, Laura Rimell, Aida Nematzadeh, Angeliki Lazaridou, Karl Moritz Hermann, Daniel Fried for helping with experiments, Chris Dyer, Sebastian Ruder and Jack Rae for their valuable feedback.

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

## APPENDIX A   HYPERPARAMETER TUNING RANGES

In all experiments, we tuned hyperparameters using Google Vizier (Golovin et al. 2017). The tuning ranges are listed in Table 4. Obviously, *mogrifier_rounds* and *mogrifier_rank* are tuned only for the Mogrifier. If $input\_embedding\_ratio \geqslant 1$, then the input/output embedding sizes and the hidden sizes are set to equal and the linear projection from the cell output into the output embeddings space is omitted. Similarly, $mogrifier\_rank \leqslant 0$ is taken to mean full rank $\mathbf{Q}^*$, $\mathbf{R}^*$ without factorization. Since Enwik8 is a much larger dataset, we don't tune *input_embedding_ratio* and specify tighter tuning ranges for dropout based on preliminary experiments (see Table 5).

Dynamic evaluation hyperparameters were tuned according to Table 6. The highest possible value for *max_time_steps*, the BPTT window size, was 20 for word, and 50 for character-level tasks. The batch size for estimating the mean squared gradients over the training data was set to 1024, gradient clipping was turned off, and the l2 penalty was set to zero.

Table 4: Hyperparameter tuning ranges for all tasks except Enwik8.

|  | Low | High | Spacing |
|---|---|---|---|
| learning_rate | 0.001 | 0.004 | log |
| input_embedding_ratio | 0.0 | 2.0 | |
| l2_penalty | 5e-6 | 1e-3 | log |
| input_dropout | 0.0 | 0.9 | |
| inter_layer_dropout | 0.0 | 0.95 | |
| state_dropout | 0.0 | 0.8 | |
| output_dropout | 0.0 | 0.95 | |
| mogrifier_rounds ($r$) | 0 | 6 | |
| mogrifier_rank ($k$) | -20 | 100 | |

Table 5: Hyperparameter tuning ranges for Enwik8.

|  | Low | High | Spacing |
|---|---|---|---|
| learning_rate | 0.001 | 0.004 | log |
| l2_penalty | 5e-6 | 1e-3 | log |
| input_dropout | 0.0 | 0.2 | |
| inter_layer_dropout | 0.0 | 0.2 | |
| state_dropout | 0.0 | 0.25 | |
| output_dropout | 0.0 | 0.25 | |
| mogrifier_rounds ($r$) | 0 | 6 | |
| mogrifier_rank ($k$) | -20 | 100 | |

Table 6: Hyperparameter tuning ranges for dynamic evaluation.

|  | Low | High | Spacing |
|---|---|---|---|
| max_time_steps | 1 | 20/50 | |
| dyneval_learning_rate | 1e-6 | 1e-3 | log |
| dyneval_decay_rate | 1e-6 | 1e-2 | log |
| dyneval_epsilon | 1e-8 | 1e-2 | log |

## APPENDIX B    HYPERPARAMETER SENSITIVITY

The parallel coordinate plots in Fig. 5 and 6, give a rough idea about hyperparameter sensitivity. The red lines correspond to hyperparameter combinations closest to the best solution found. To find the closest combinations, we restricted the range for each hyperparameter separately to about 15% of its entire tuning range.

For both the LSTM and the Mogrifier, the results are at most 1.2 perplexity points off the best result, so our results are somewhat insensitive to jitter in the hyperparameters. Still, in this setup, grid search would require orders of magnitude more trials to find comparable solutions.

On the other hand, the tuner does take advantage of the stochasticity of training, and repeated runs with the same parameters may be give slightly worse results. To gauge the extent of this effect, on PTB we estimated the standard deviation in reruns of the LSTM with the best hyperparameters to be about 0.2 perplexity points, but the mean was about 0.7 perplexity points off the result produced with the weights saved in best tuning run.

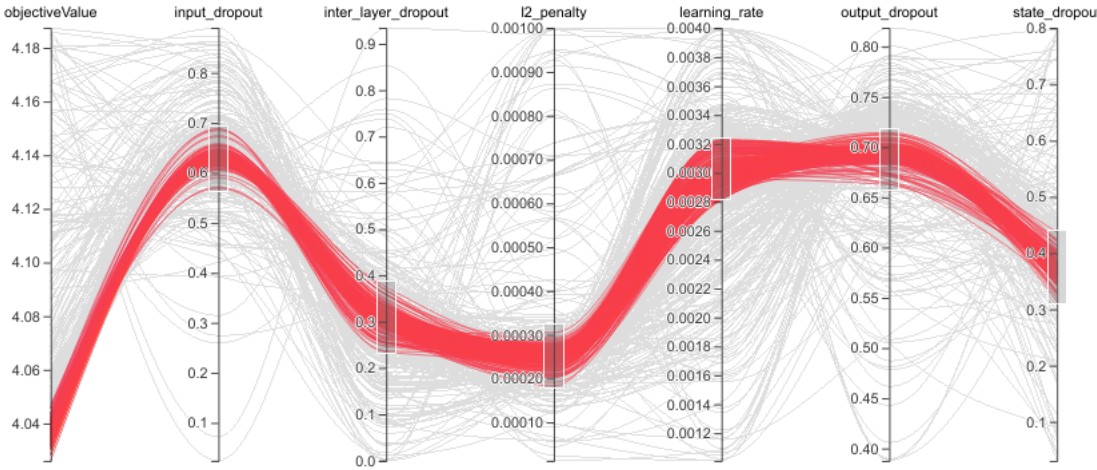

Figure 5: Average per-word validation cross-entropies for hyperparameter combinations in the neighbourhood of the best solution for a 2-layer LSTM with 24M weights on the Penn Treebank dataset.

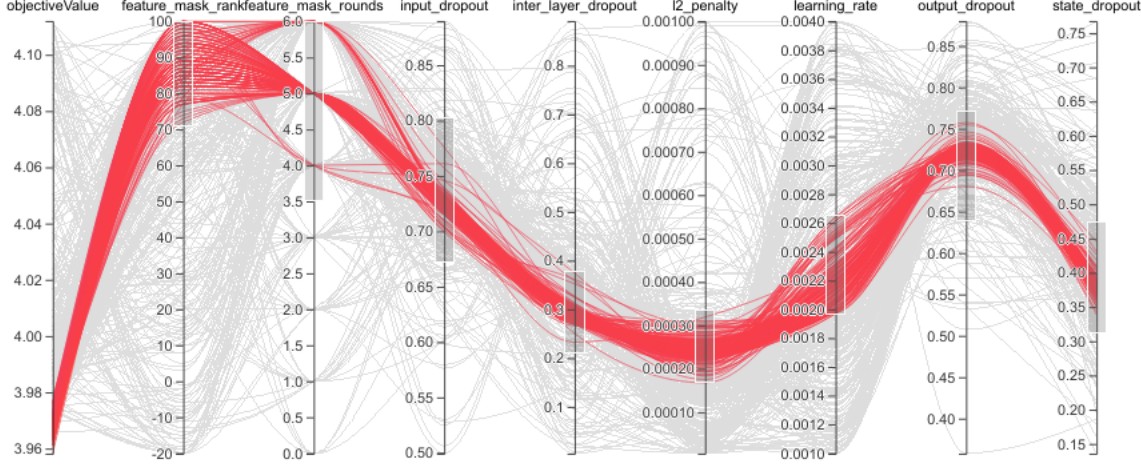

Figure 6: Average per-word validation cross-entropies for hyperparameter combinations in the neighbourhood of the best solution for a 2-layer Mogrifier LSTM with 24M weights on the Penn Treebank dataset. *feature_mask_rank* and *feature_mask_rounds* are aliases for *mogrifier_rank* and *mogrifier_rounds*
.

