# OpenReview forum: "Mogrifier LSTM"
_ICLR.cc/2020/Conference — Accept (Talk)_

### Official Review · AnonReviewer2 · 2019-10-22
**Official Blind Review #2**

**Rating:** 8

**Review:**

Summary:

The paper proposes a novel LSTM architecture that adds several gating mechanism that gates the hidden state and inputs in between the LSTM update. The proposed model shows superior performance on smallish datasets including PTB,  Enwick8 and NWC.

Comments on the paper:

1. The paper proposes an interesting architecture and it seems to show significant improvement in terms of performance for some language datasets.

2. The paper is very well written, the motivation and formulation is clear. There are many analysis to understand the model (the strength and weaknesses).

3.. One thing is that since this could take into account more context,  it seems that this model could potentially generate language / tokens with longer time dependencies. I wonder if the authors have performed any experiments on this and if they have seen any improvements on that front.

4. Also, I am curious about the generalization ability of the model. Could the authors train the model on shorter sequences and test for generation with longer sequences and see how this compares with baseline models.

5. The model seems to be related to Adaptive Computation Time (ACT) from Gaves et al. it would be nice to compare to the ACT.

6. Another slight improvement in writing could be to hightlight the intuition (conclusion in page 8) at the beginning of the paper, this could help in better understanding the motivation of the paper.


Minor comments on the paper,

1. The link and the self-citations on page 4 are does not seem to be valid links and citations.

Overall, a well-written paper, extensive analysis and good experimental result.

**Experience Assessment:**

I have published one or two papers in this area.

**Review Assessment: Checking Correctness Of Derivations And Theory:**

I assessed the sensibility of the derivations and theory.

**Review Assessment: Checking Correctness Of Experiments:**

I assessed the sensibility of the experiments.

**Review Assessment: Thoroughness In Paper Reading:**

I read the paper at least twice and used my best judgement in assessing the paper.

---

> ### Author Response · Authors · 2019-11-07
> **Re: Official Blind Review #2**
>
> We thank Reviewer #2 for their comments.
>
> > 3. One thing is that since this could take into account more
> >    context, it seems that this model could potentially generate
> >    language / tokens with longer time dependencies. I wonder if the
> >    authors have performed any experiments on this and if they have
> >    seen any improvements on that front.
> >
> > 4. Also, I am curious about the generalization ability of the model.
> >    Could the authors train the model on shorter sequences and test
> >    for generation with longer sequences and see how this compares
> >    with baseline models.
>
> The only relevant bit in the paper is this: "Hypothesis: the benefit
> is in handling long-range dependencies better. Experiments in the
> episodic setting (i.e. sentence-level language modelling) exhibited
> the same gap as the non-episodic ones."
>
> Clearly, this does not exactly rule out the possibility of long-range
> vs short-range being a factor, but at the same time it sounds somewhat
> unlikely to observe improvements of the same magnitude in per-sentence
> language modelling.
>
> Evaluating generated text can be rather subjective though, so we
> refrained from that.
>
> > 5. The model seems to be related to Adaptive Computation Time (ACT)
> >    from Graves et al. it would be nice to compare to the ACT.
>
> We agree that ACT could be related. The main difference is that the
> mogrifier in its current version performs a fixed number of processing
> steps.
>
> > 6. Another slight improvement in writing could be to hightlight the
> >    intuition (conclusion in page 8) at the beginning of the paper,
> >    this could help in better understanding the motivation of the
> >    paper.
>
> Thank you for the suggestion. We'll consider this.

---

### Official Review · AnonReviewer3 · 2019-10-23
**Official Blind Review #3**

**Rating:** 8

**Review:**

Summary:
This paper tackles the problem of context modelling within recurrent neural networks (RNNs). The authors propose an interdependent gating mechanism that enriches the coupling between inputs and hidden states. For an input x_0 and hidden state h_0; h_0 gates x_0 to create x_1; x_1 then gates h_0 to create h_1; this cyclical gating operation is applied for several rounds and it's output is fed into a recurrent neural network. For the next time-step, this process is repeated, with h_0 as the final h obtained after the final round of gating in the previous time-step. This results in the RNN processing a more contextualized version of the input tokens x.

Main Contributions:
1. A simple pre-processing step that contextualizes inputs for recurrent neural networks and significantly improves performance.
2. An extensive evaluation of the proposed technique against previous works and on all relevant datasets.

Pros:
The paper is very well-written and clear. It motivates and explores the questions and issues surrounding this topic very well.

Cons:
It would be good to see how this performance translates to other RNN architectures such as GRUs.


Final notes:
This paper raises many interesting question:
- What is the really going on with the gating mechanism?
The authors explore this question but the jury is still out on exactly what is going on here.
- "Mogrification" as a general preprocessing step: could it also improve performance for transformer models?
- Are there better ways to preprocess and gate the RNN inputs?

--------

Review Decision:
It is clear, well motivated, well written and represents a concrete contribution to the language modelling literature. Furthermore, most claims made are substantiated via thorough experimentation. Lastly, this work demonstrates that rather than relying on data and model scaling to improve performance; there is alot left to be done in tackling language modelling on smaller scale datasets.

**Experience Assessment:**

I have published in this field for several years.

**Review Assessment: Checking Correctness Of Derivations And Theory:**

I carefully checked the derivations and theory.

**Review Assessment: Checking Correctness Of Experiments:**

I carefully checked the experiments.

**Review Assessment: Thoroughness In Paper Reading:**

I read the paper thoroughly.

---

> ### Author Response · Authors · 2019-11-07
> **Re: Official Blind Review #3**
>
> We thank Reviewer #3 for their comments.
>
> - "Mogrification" as a general preprocessing step: could it also
>   improve performance for transformer models?
>
> Possibly, but it would be quite surprising, as attention might very
> well be able to express similar transformations.
>
> As to whether there better ways to preprocess and gate the RNN inputs,
> we are sure that the answer is yes. More generally, we believe that
> our neural models lack the necessary biases perform well in a
> data-efficient manner. Large datasets can alleviate the problem, but
> may not be able to solve it.

---

### Official Review · AnonReviewer1 · 2019-10-25
**Official Blind Review #1**

**Rating:** 6

**Review:**

I have read the authors' response. Their points regarding baseline comparisons are sensible in that there isn't a reason to expect the observations to *not* generalization to other datasets. It is odd that mLSTM is outperformed by LSTM in Table 3, but as the authors note in section 4.2 this may be due to instability of mLSTM during training. The results in the paper demonstrate significant improvement over LSTM, and while there are not as many baseline comparison to similar models as I would have liked to see, the quality of this work is sufficiently high that this is not a fatal flaw. In light of the author response and other reviews, I am revising my rating to 6: Weak Accept.

=====

This paper proposes a modification of LSTM networks in the context of language modeling called Mogrifier LSTM. Ordinary LSTMs are defined as recurrent operations on the current input, previous hidden state, and previous cell state. The proposed Mogrifier LSTM utilizes the same recurrent unit as the LSTM, but the input and previous hidden state are updated with several rounds of mutual gating. In each round, the input  is multiplied elementwise by a gate computed as a function of the hidden state (or vice versa). The authors experiment on word-level and character-level modeling and compare their Mogrifier LSTMs to several state-of-the-art approaches. They also conduct an ablation study to show the effect of various design choices and hyperparameters and experiments on a reverse copy task.

Specific contributions include:
* Proposal of a novel approach for modulating inputs to a recurrent unit by mutual gating.
* Experiments demonstrating strong performance on a number of language modeling tasks.

The paper in its current state is borderline, leaning towards weak reject. Points in favor of acceptance include the high clarity of writing, good experiments of the proposed model, and a discussion of possible reasons for why the mogrification operation works well. The main shortcoming of the paper is experimental comparison to baselines.

The authors were able to train baseline LSTMs to high levels of performance (presumably due to tuning of hyperparameters) and then demonstrate that Mogrifier LSTMs improve upon LSTMs significantly. This is perhaps not entirely surprising, because the hyperparameter range of the Mogrifier LSTM includes zero rounds of updates, which would render it identical to the baseline LSTM. Therefore, if the hyperparameters are tuned sufficiently well, the performance of the Mogrifier LSTM should be at least as good as the LSTM. What the experiments do not show is that the proposed mogrification outperforms other forms of multiplicative interaction and/or gating. The closest that the authors come to this is the single validation perplexity of the Multiplicative LSTM in Table 3. If thorough hyperparameter tuning is applied to the Multiplicative LSTM or the approaches of Wu et al. (2016) and/or Sutskever et al. (2011), does the Mogrifier LSTM still outperform them?

Other than this critical issue of baseline comparison, the experiments are quite informative. The ablation study showing the effect of different design decisions and the hyperparameter visualiztion in Appendix B are particularly useful. The mogrification operation is described precisely enough for other researchers to implement and the arguments made in 4.4 are compelling.

Question for the authors:
* Some qualitative analysis of the learned mogrification operation would be helpful for understanding the nature of the modulation. For example, how do the predictions change depending on the modulation? If x is modulated by different hidden states, is there a noticeable effect on the output?
* Did you experiment with other forms of modulation before arriving upon the mogrification formulation? There are some naive approaches such as concatenating the hidden state to the input and applying a nonlinear layer, or predicting affine parameters for the input as a function of the hidden state in the style of FiLM [1]. Are there obvious shortcomings in these naive approaches that mogrification handles gracefully?

[1] Perez, E., Strub, F., De Vries, H., Dumoulin, V. and Courville, A., 2018, April. Film: Visual reasoning with a general conditioning layer. In Thirty-Second AAAI Conference on Artificial Intelligence.

**Experience Assessment:**

I do not know much about this area.

**Review Assessment: Checking Correctness Of Derivations And Theory:**

I assessed the sensibility of the derivations and theory.

**Review Assessment: Checking Correctness Of Experiments:**

I assessed the sensibility of the experiments.

**Review Assessment: Thoroughness In Paper Reading:**

I read the paper at least twice and used my best judgement in assessing the paper.

---

> ### Author Response · Authors · 2019-11-07
> **Re: Official Blind Review #1**
>
> We thank Reviewer #1 for the critical but thoughtful review. We try to
> address the issues brought up below.
>
> > The authors were able to train baseline LSTMs to high levels of
> > performance (presumably due to tuning of hyperparameters) and then
> > demonstrate that Mogrifier LSTMs improve upon LSTMs significantly.
> > This is perhaps not entirely surprising, because the hyperparameter
> > range of the Mogrifier LSTM includes zero rounds of updates, which
> > would render it identical to the baseline LSTM. Therefore, if the
> > hyperparameters are tuned sufficiently well, the performance of the
> > Mogrifier LSTM should be at least as good as the LSTM
>
> It is indeed unsurprising that the Mogrifier is not worse than the
> LSTM since it includes the LSTM as a special case. But in Figure 3
> where perplexity is plotted as the function of rounds, the setting
> that corresponds to the LSTM (rounds=0) is clearly the worst and the
> gap is very significant.
>
> > What the experiments do not show is that the proposed mogrification
> > outperforms other forms of multiplicative interaction and/or gating.
> > The closest that the authors come to this is the single validation
> > perplexity of the Multiplicative LSTM in Table 3. If thorough
> > hyperparameter tuning is applied to the Multiplicative LSTM or the
> > approaches of Wu et al. (2016) and/or Sutskever et al. (2011), does
> > the Mogrifier LSTM still outperform them?
>
> We agree that having these baselines would strengthen the contribution
> and help position our work more precisely in their context. Due to
> time and resource constraints, we focussed on the most similar model,
> the mLSTM, and evaluated it on PTB (which has been predictive of
> performance on other tasks in our experience) using the same
> hyperparameter tuning methodology as everywhere else in the paper, the
> only exceptions being a shortened schedule and small BPTT window size.
> These concessions to practicality make results slightly worse (2-3
> perplexity points), but there is little reason to believe they benefit
> one model or the other. And if the mLSTM were more similar to the
> mogrifier than we'd like, we should see that in these experiments. As
> it is, what we found is that the mLSTM does not improve on the
> baseline LSTM while the Mogrifier does.
>
> > * Some qualitative analysis of the learned mogrification operation
> >   would be helpful for understanding the nature of the modulation. For
> >   example, how do the predictions change depending on the modulation? If
> >   x is modulated by different hidden states, is there a noticeable
> >   effect on the output?
>
> Obviously, in terms of perplexity there is a noticable effect. In
> terms of statistics of the modulated vs unmoodulated input vectors, we
> do not have the data. The closest we have in the paper is that "the
> means of the standard LSTM gates in the Mogrifier were very close
> between the two models but their variance was smaller in the
> Mogrifier".
>
> > * Did you experiment with other forms of modulation before arriving
> >   upon the mogrification formulation? There are some naive
> >   approaches such as concatenating the hidden state to the input and
> >   applying a nonlinear layer, or predicting affine parameters for
> >   the input as a function of the hidden state in the style of FiLM
> >   [1]. Are there obvious shortcomings in these naive approaches that
> >   mogrification handles gracefully?
>
> We tried concatenation of hidden state and input and saw no benefit
> compared to the Mogrifier to offset the significantly higher number of
> parameters. FiLM sounds similar to one round mogrifier without a
> non-linearity. As to obvious shortcomings to these methods, we do not
> know of any. Probably we would need to understand a mogrifier much
> better to answer that question.

---

### Decision · Program_Chairs · 2019-12-19

**Decision:**

Accept (Talk)

**Comment:**

This paper presents a new twist on the typical LSTM that applies several rounds of gating on the history and input, with the end result that the LSTM's transition function is effectively context-dependent. The performance of the model is illustrated on several datasets.

In general, the reviews were positive, with one score being upgraded during the rebuttal period. One of the reviewers complained that the baselines were not adequate, but in the end conceded that the results were still worthy of publication.

One reviewer argued very hard for the acceptance of this paper "Papers that are as clear and informative as this one are few and far between. ... As such, I vehemently argue in favor of this paper being accepted to ICLR."